# Circuit implementation of a four-dimensional topological insulator

You Wang[1], Hannah M. Price [2], Baile Zhang [1,3✉] & Y. D. Chong[1,3✉]

The classification of topological insulators predicts the existence of high-dimensional topological phases that cannot occur in real materials, as these are limited to three or fewer spatial dimensions. We use electric circuits to experimentally implement a four-dimensional (4D) topological lattice. The lattice dimensionality is established by circuit connections, and not by mapping to a lower-dimensional system. On the lattice's three-dimensional surface, we observe topological surface states that are associated with a nonzero second Chern number but vanishing first Chern numbers. The 4D lattice belongs to symmetry class AI, which refers to time-reversal-invariant and spinless systems with no special spatial symmetry. Class AI is topologically trivial in one to three spatial dimensions, so 4D is the lowest possible dimension for achieving a topological insulator in this class. This work paves the way to the use of electric circuits for exploring high-dimensional topological models.

[1] Division of Physics and Applied Physics, School of Physical and Mathematical Sciences, Nanyang Technological University, Singapore 637371, Singapore. [2] School of Physics and Astronomy, University of Birmingham, Edgbaston, Birmingham B15 2TT, UK. [3] Centre for Disruptive Photonic Technologies, Nanyang Technological University, Singapore 637371, Singapore. ✉email: blzhang@ntu.edu.sg; yidong@ntu.edu.sg

Topological insulators are materials that are insulating in the bulk but host surface states protected by nontrivial topological features of their bulk bandstructures[1,2]. They are classified according to symmetry and dimensionality[3–7], with each class having distinct and interesting properties. The celebrated two-dimensional Quantum Hall (2DQH) phase[8], for instance, has topological edge states that travel unidirectionally on the one-dimensional (1D) edge, whereas three-dimensional (3D) topological insulators based on spin-orbit coupling have surface states that act like massless 2D Dirac particles. The classification of topological insulators contains hypothetical high-dimensional phases[3] that cannot be realised with real materials, since electrons only move in one, two, or three spatial dimensions. These include several types of four-dimensional Quantum Hall (4DQH) phases, which are characterised by a 4D topological invariant called the second Chern number and exhibit a much richer phenomenology than the 2DQH phase[9–12]. In recent years, topological phases have been implemented in a range of engineered systems including cold atom lattices[13], photonic structures[14], acoustic and mechanical resonators[15,16], and electric circuits[17–28]. Some of these platforms can realise lattices that are hard to achieve in real materials, raising the intriguing prospect of using them to create high-dimensional topological insulators. Although there have been demonstrations of topological pumps that map 4D topological lattice states onto lower-dimensional systems[29–32], there has been no experimental realisation of a 4D topological insulator with protected surface states on a 3D surface.

Here, we describe the implementation of a 4DQH phase using electric circuits to access higher dimensions. Since electric circuits are defined in terms of lumped (discrete) elements and their interconnections, lattices with genuine high-dimensional structure can be explicitly constructed by applying the appropriate connections[33–35]. In this way, we experimentally implement a 4D lattice hosting the first realisation of a Class AI topological insulator[5,6], which has no counterpart in three or fewer spatial dimensions.

In the symmetry-based classification of topological phases[3–7], Class AI includes time-reversal ($T$) symmetric, spinless systems that are not protected by any special spatial symmetries. Whereas the 2DQH phase is tied to nontrivial values of the first Chern number, which requires T-breaking[36], 4DQH phases rely on the second Chern number, which does not[9–12]. Even though the Class AI conditions are ubiquitous[13,14], the class is topologically trivial in one to three dimensions[3–7]. Hence, realising a Class AI topological insulator requires going to at least 4D. We focus on a theoretical 4D lattice model recently developed by one of the authors[37], which exhibits a nonzero second Chern number with vanishing first Chern numbers. Hence, we obtain the first observations of topological surface states that are intrinsically tied to 4D band topology, with no connection to lower-dimensional topological invariants.

The present approach, based on circuit connections, is distinct from other recently-investigated methods for accessing higher-dimensional models. One of the alternatives involves manipulating internal degrees of freedom, such as oscillator modes, to act as synthetic dimensions[38–52]. Although there have been theoretical proposals for using synthetic dimensions to build 4D topological lattices[40,43], all experiments so far have been limited to 1D and 2D[51]. Another approach involves adiabatic topological pumping schemes, which map high-dimensional models onto lower-dimensional setups by replacing spatial degrees of freedom with tunable parameters[29–32]. As mentioned above, 2D topological pumps based on cold atoms and photonics have recently been used to explore Class A (T-broken) 4DQH systems[30,53,54]. However, topological pumps have the drawback of being inherently limited to probing specific quasi-static solutions of a high-dimensional system, without realising a genuinely high-dimensional lattice. Moreover, in those experiments the second Chern number in 4D is not truly independent of the first Chern numbers in 2D, which are nonzero.

Our 4D lattice is implemented using electric circuits with carefully chosen capacitive and inductive connections. The lattice model has two topologically distinct phases: a 4DQH phase and a conventional (i.e. topologically trivial) 4D band insulator, with the choice of phase governed by a parameter $m$ that maps to certain combinations of capacitances and inductances. Using impedance measurements that are equivalent to finding the local density of states (LDOS), we show that the 4DQH phase hosts surface states on the 3D surface, while the conventional insulator phase has only bulk states. By varying the driving frequency, we show that the topological surface states span a frequency range corresponding to a bulk bandgap, as predicted by theory. Our experimental results also agree well with circuit simulations. This work demonstrates that electric circuits are a flexible and practical way to realise higher-dimensional lattices, paving the way for the exploration of other previously-inaccessible topological phases.

## Results

**4DQH model and circuit realisation.** The 4D lattice model is shown schematically in Fig. 1a. The spatial coordinates are denoted $x$, $y$, $z$, and $w$. The lattice contains four sublattices labelled A, B, C and D, with sites connected by real nearest neighbour hoppings $\pm J$. The four bands host two pairs of Dirac points in the Brillouin zone; each pair is the time-reversed counterpart of the other. To control the pairs separately, long-range hoppings with amplitudes $\pm J'$ and $\pm J''$ are added within the $x$–$z$ plane (these hoppings are omitted from Fig. 1a for clarity, but are shown in Fig. 1c). Upon adding mass $+m$ to the A and B sites, and $-m$ to the C and D sites, the Dirac masses for the different Dirac point pairs close at $m = J' - 2J''$ and $m = J'' - 2J'$. These gap closings are topological transitions, such that, for $J'' = -J'$, the second Chern number of the lower bands is $-2$ (nontrivial) if $|m| < 3|J'|$. Since $T$ is unbroken, the first Chern number is always zero, so the model exhibits QH behaviour stemming purely from the second Chern number[37]. For further details about the model, see Supplementary Note 1.

We take $J = 1$ and $J' = -J'' = 2$, so that the topological transition of the bulk lattice occurs at $m = \pm 6$. We target a finite 4D lattice with six sites along the $x$ and $z$ directions, and two sites along $y$ and $w$. To mitigate finite-size effects, periodic boundary conditions are applied in $y$ and $w$ using nearest neighbour type connections between opposite ends of the lattice. This corresponds to sampling at $k_y = k_w = 0$ in momentum space, where the gap closing occurs during the topological transition (see Supplementary Note 1). Regardless of these periodic boundary conditions, the spatial dimensionality established by the connectivity of the lattice sites is 4D[33]. The lattice has a total of 144 sites, of which we consider 16 to be bulk sites (defined as being more than two sites away from a surface) and 128 to be surface sites.

The finite 4D lattice is implemented with a set of connected printed circuit boards, shown in Fig. 1b. Each site $i$ of the tight-binding model maps to a node on the circuit, and the mass term maps to a circuit component of conductance $-D_{ii}$ connecting the node to ground. Each hopping $J_{ij}$ between sites $i$ and $j$ maps to a circuit element of conductance $D_{ij}$ connecting the nodes. We add extra grounding components with conductance $D'_{ii}$ in parallel with $-D_{ii}$. If an external AC current $I_i$ flows into each node $i$ at frequency $f$, and $V_i$ is the complex AC voltage on that node,

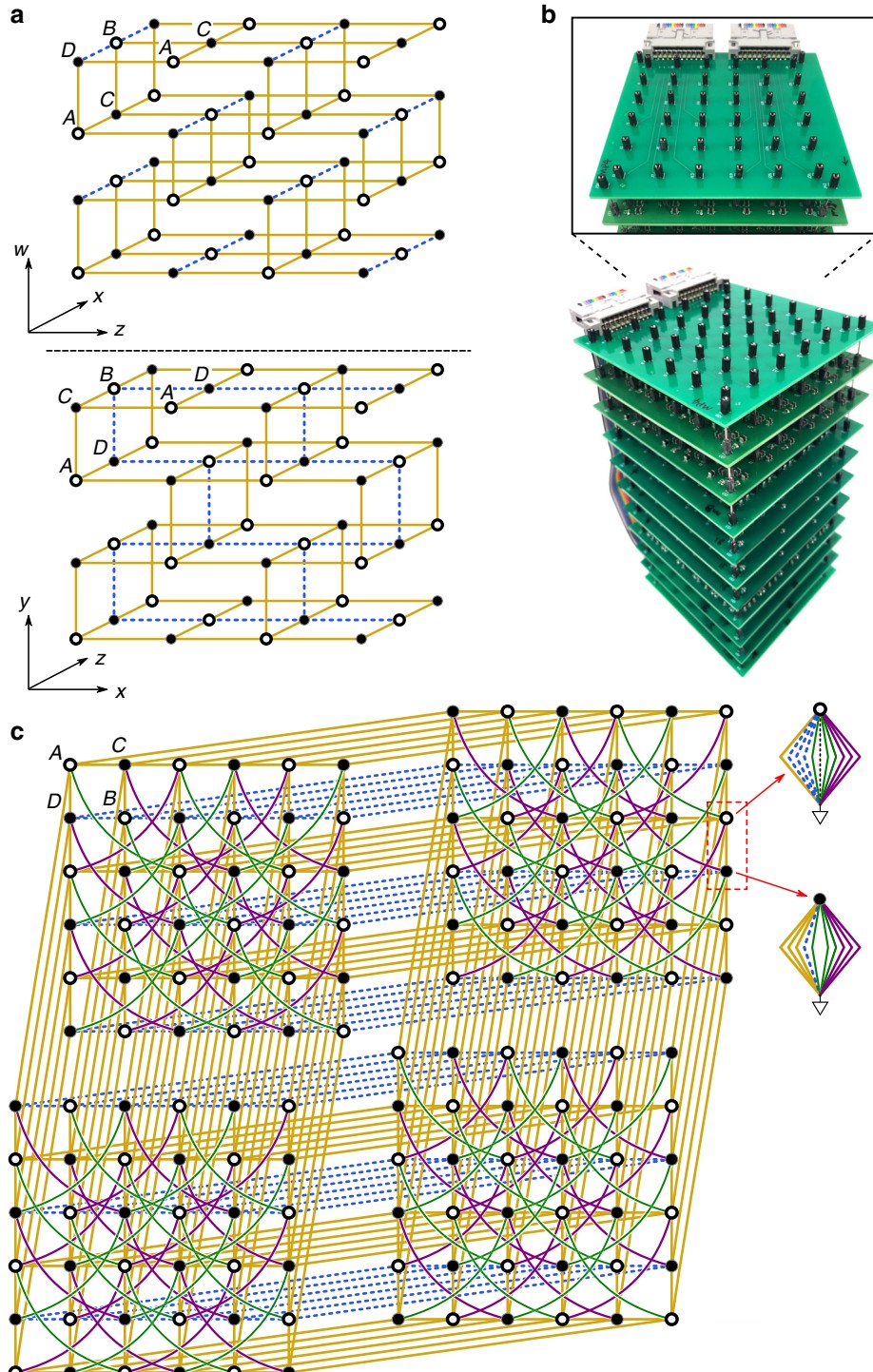

**Fig. 1 Model of the 4D Quantum Hall lattice and its circuit implementation. a** Schematic of the 4D tight-binding model. Each unit cell consists of four sites labelled A–D. Hollow and filled circles respectively denote positive ($m$) and negative ($-m$) on-site masses, while yellow solid lines and blue dashes respectively denote positive ($J$) and negative ($-J$) hoppings. **b** Photographs of the circuit. **c** Schematic of the circuit; positive (negative) masses are realised by capacitors (inductors) connecting the sites to ground, and hoppings are realised using capacitors or inductors connecting different sites. The components shown here are $C_0 = 1$ nF (yellow lines), $L_0 = 2$ mH (blue dashes), $C' = 2$ nF (purple curves), $L' = 1$ mH (green curves), and $C_m = 2mC_0$ (grey dashes). Each site is grounded by a set of additional circuit components (see Supplementary Note 2); for clarity, only the grounding components for two sites (in the dashed red box) are depicted.

Kirchhoff's law states that

$$I_i = (-D_{ii} + D'_{ii})V_i + \sum_j D_{ij}(V_i - V_j).$$ (1)

We define $D_{ij}(f) = i\alpha H_{ij}(f)$, where $\alpha$ is a positive real constant. Then capacitances (inductances) correspond to positive (negative) real values of $H_{ij}$. We require that at a reference working frequency $f = f_0$, the values of $H_{ij}(f_0)$ match the desired tight-

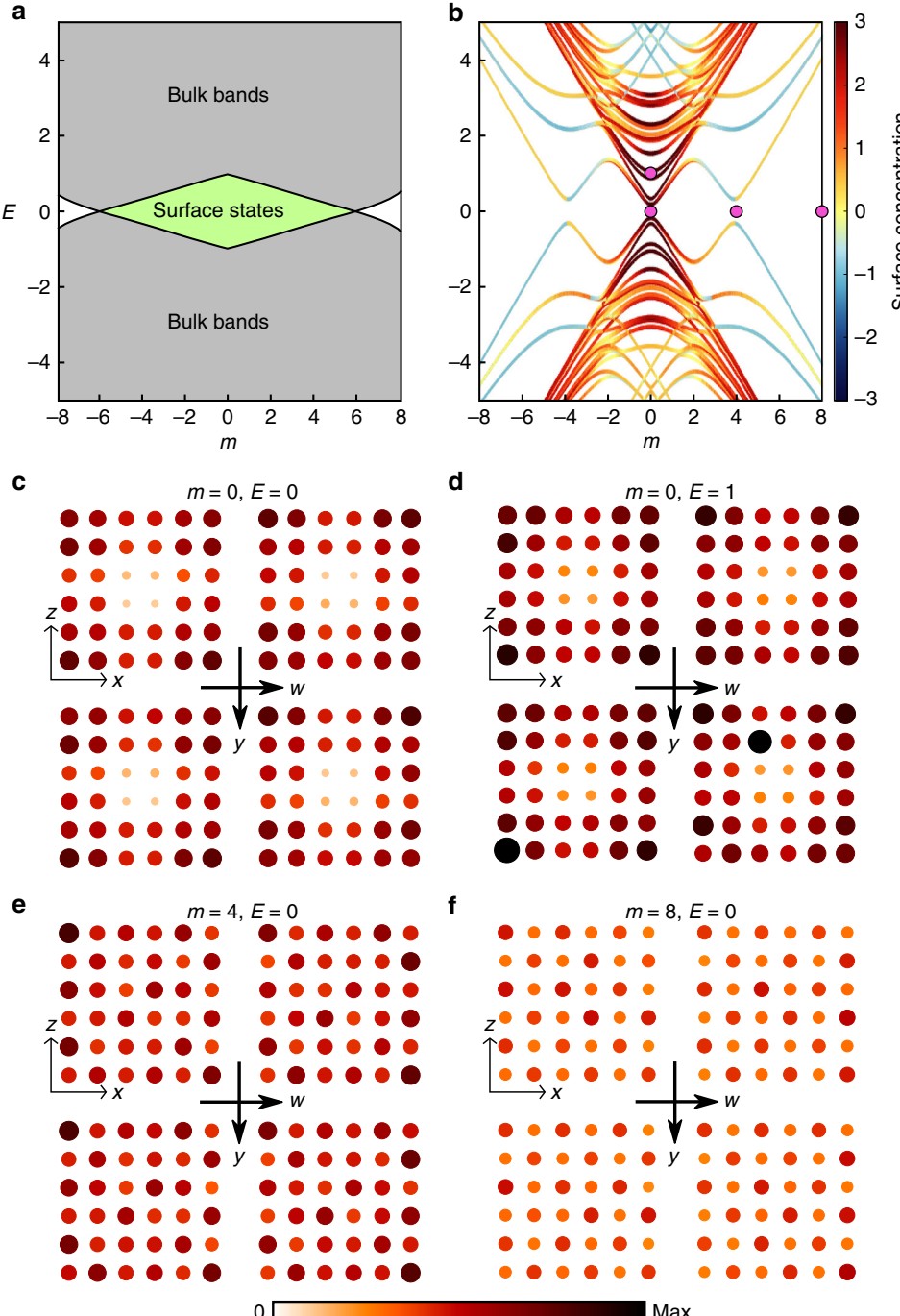

**Fig. 2 Band diagram and local density of states of the 4D lattice. a** Calculated band diagram for the infinite 4D lattice. The bulk bands are shown in grey. For $|m| < 6$, there is a bandgap associated with nontrivial second Chern number, accompanied by topological surface states (shaded green). For $|m| > 6$, the bandgap is trivial. **b** Calculated band diagram for 144-site lattice with periodic boundary conditions along $y$ and $w$. Colours indicate the degree of surface concentration of the energy states, as defined Eq. (5). Due to finite-size effects, surface states occur at $|m| \lesssim 2$ and the gap closing is shifted to $|m| \approx 4$. The parameters corresponding to subplots **c–f** are indicated with pink dots. **c–f** Experimentally-obtained LDOS maps for different $m$ and $E$, measured at working frequency $f = f_0$. Surface states are observed in **c**, **d** consistent with theoretical predictions.

binding lattice Hamiltonian (see "Methods"). We then tune $D'$ so that for $f = f_0$,

$$D'_{ii} + \sum_{j \neq i} D_{ij} = i\alpha E, \qquad (2)$$

for some target energy $E$; the required value of $D'$ depends on the

$m$ parameter. Equation (1) now becomes

$$I_i \equiv \sum_j L_{ij} V_j = -i\alpha \sum_j [H_{ij}(f) - E\,\delta_{ij}] V_j(f), \qquad (3)$$

where $L_{ij}$ are the components of the circuit Laplacian $L$. The

impedance between node $r$ and ground is

$$V_r = \sum_j (L^{-1})_{rj} I_j = Z_r I_r. \tag{4}$$

It can be shown that $\mathrm{Re}[Z_r(f_0)]$ is, up to a scale factor, the LDOS of the target lattice at energy $E$ (see "Methods"). For further details about the circuit analysis, see Supplementary Note 2.

**Experimental results**. Figure 2a shows the band diagram of the infinite bulk tight-binding model as a function of the mass detuning parameter $m$. For $|m| < 6$, the system is in a 4DQH phase, with a topologically nontrivial bandgap centred at $E = 0$, which hosts topological surface states.

The band diagram for the 144-site tight-binding model is shown in Fig. 2b. The colours of the curves indicate the degree to which each eigenstate is concentrated on the surface, as defined by

$$\ln\left[ \langle|\psi(r)|\rangle_{\mathrm{surf.}} / \langle|\psi(r)|\rangle_{\mathrm{bulk}} \right], \tag{5}$$

where $\psi(r)$ denotes the energy eigenfunction, whose magnitudes are averaged over either surface or bulk sites. Due to the finite lattice size, both the bulk and surface spectrum are split into subbands. The closing of the bulk gap is shifted to $|m| \approx 4$, and the surface states occur most prominently at small values of $E$ and $|m|$.

We now fabricate a set of circuits with parameters $m \in \{0, 1, …, 8\}$ and $E \in \{0, 1\}$. Figure 2c–f shows the measured LDOS (at $f = f_0$) for four representative samples. From the experimental data, we see that the surface LDOS is high and the bulk LDOS is low when in the topologically nontrivial bandgap (Fig. 2c, d). For $E = 0$, $m = 4$, which corresponds roughly to the gap-closing point, there is no significant difference between the surface and bulk LDOS. For $E = 0$, $m = 8$, the LDOS on all sites is low, consistent with being in a topologically trivial bandgap. These results also agree well with circuit simulations (see Supplementary Note 3). The robustness of the surface states, a feature imparted by topological protection, can be inferred from the fact that each individual circuit component has up to 10% deviation in its capacitance or inductance (see "Methods"). We emphasise that the surface states cannot be explained by the first Chern numbers, which are necessary zero since the circuit design is $T$ symmetric.

To confirm that the discrepancy between Fig. 2a and b is just a finite-size effect, Fig. 3 shows calculated band edges (i.e. the pair of eigenvalues closest to $E = 0$) for a series of lattices with 6, 8, 10, 14, 20, and 50 sites along both $x$ and $z$ (the lattices are kept two sites wide along $y$ and $w$, with periodic boundary conditions). The colours indicate whether the eigenstate is concentrated on the surface (red) or in the bulk (blue). As the size of the lattice increases in $x$ and $z$, the eigenvalues at large $m$ (in the conventional insulator regime) approach the predicted bulk band edges, while the eigenvalues in the topological insulator regime spread over a larger range of $m$ corresponding to the topologically nontrivial gap.

To quantify the difference between the 4DQH and conventional insulator phases, we examine the ratio of the mean LDOS on surface sites to the mean LDOS on bulk sites, for different values of the mass detuning parameter $m$ (Fig. 4a). The ratio is derived from experimental measurements performed at $f = f_0$, corresponding to $E = 0$; with increasing $m$, it decreases sharply from around 4.5 in the 4DQH regime to around 1 in the conventional insulator regime. Circuit simulations produce results in agreement with the experimental data (Fig. 4f).

The frequency dependence of the circuit impedance is also consistent with the spectral features of a topological insulator at small values of $m$. Figure 4b–e plots the experimentally-obtained

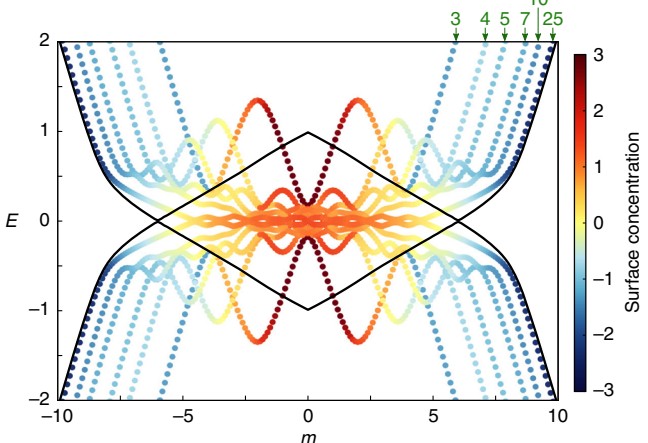

**Fig. 3 Band diagrams for lattices of different sizes.** Eigenvalue pairs closest to $E = 0$ for lattices with 6, 8, 10, 14, 20, and 50 sites along the $x$ and $z$ directions (lengths labelled in the upper right corner), and two sites along $y$ and $w$ (with periodic boundary conditions). Black curves show the bulk band edges.

frequency dependence of the LDOS measure $\mathrm{Re}[Z_r]$, averaged over surface or bulk sites. To interpret these results, recall that the impedance measurements probe the response at fixed energy (in this case, $E = 0$) of an effective Hamiltonian $H(f)$ that depends parametrically on the frequency $f$ [Eq. (6)], and matches the target tight-binding model at $f = f_0$. For $m = 0$ (Fig. 4b), the circuit exhibits a strong edge response and suppressed bulk response at $f = f_0$, consistent with the fact that $H(f_0)$ has a topologically nontrivial gap at $E = 0$. For $f \neq f_0$, the effective Hamiltonian $H(f)$ deviates from the target model (e.g. the positive and negative hoppings become unequal in magnitude, lifting the band degeneracy), but remains in Class AI. So long as the gap remains open, $H(f)$ must possess a topologically nontrivial gap at $E = 0$ associated with the same second Chern number. The signatures of the topological bandgap persist as $m$ is slightly increased (Fig. 4c); upon further increasing $m$, the bulk gap closes and thereafter the surface and bulk LDOS measures exhibit no notable frequency dependent features (Fig. 4d, e). These experimental results are in good agreement with simulations (Fig. 4g–j).

## Discussion
We have used electric circuits to implement a 4D lattice hosting a 4D Quantum Hall phase. This is the first experimental demonstration of a topological lattice with a 4D structure, and of a Class AI topological insulator. This is also the first experimental exploration of a 4DQH model with nontrivial second Chern number but trivial first Chern numbers. Using impedance measurements, we have demonstrated that the LDOS on the 3D surface is enhanced in the 4DQH phase, due to the presence of topological surface states, and that the enhanced surface response spans the frequency range of the bulk bandgap. The gap-closing associated with a topological phase transition is clearly observed, despite being shifted by finite-size effects. In future work, it is desirable to find ways to probe the detailed features of the 3D surface states, which are predicted to be two robust isolated Weyl points of the same chirality, a situation that does not occur in lower-dimensional topological models[37]. The successful implementation of 4D lattices of very substantial size (144 sites) shows that electric circuits are an excellent platform for exploring exotic band topological effects, and a promising alternative to the synthetic dimensions approach to realising higher-dimensional lattices[51].

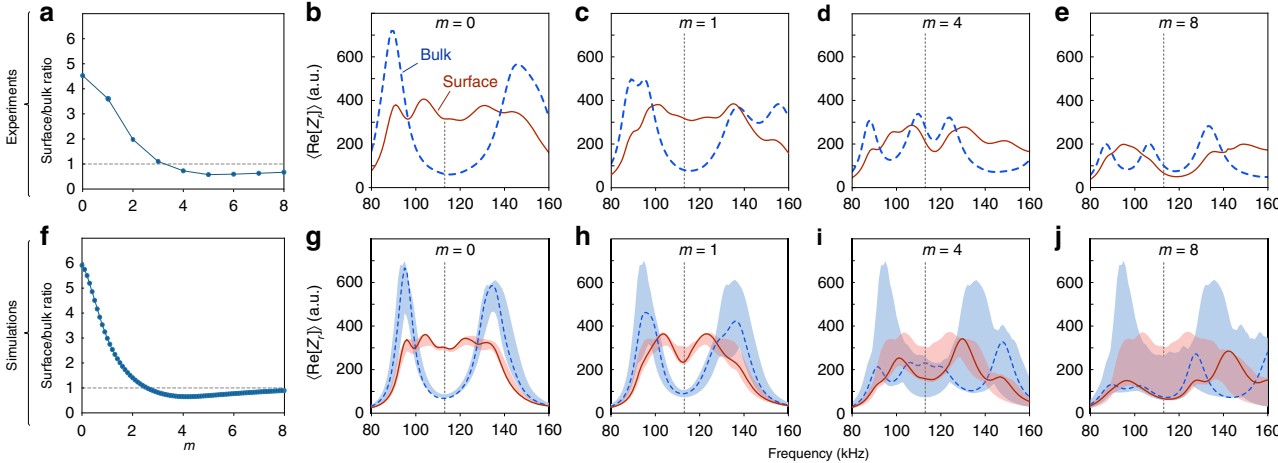

**Fig. 4 Comparison of bulk and surface contributions to the LDOS. a** Ratio of surface to bulk LDOS, measured at $f = f_0$, versus $m$. **b-e** Mean values of the LDOS measure $\mathrm{Re}[Z_r]$ on surface and bulk sites, versus working frequency $f$. For these subplots, measurements were only taken over sites in the 2D plane $(y, w) = (1, 0)$. The reference working frequency $f_0$ (corresponding to $E = 0$) is indicated by the vertical dotted line. For small $m$, we observe an elevated surface LDOS measure over a range of frequencies coincident with a bulk gap. Upon increasing $m$, the gap closes. **f-j** The corresponding circuit simulation results, obtained using the same circuit parameters and with manufacturer-supplied estimates for the capacitor and inductor resistances. The results in **f**, and the solid curves and dashes in **g-j**, assume no disorder in the circuit components. The red and blue areas in **g-j** indicate the range of impedances assuming 10% variation in individual capacitances and inductances, over 50 independent disorder realisations.

While this work was being done, we became aware of related theoretical proposals to use circuits to realise high-dimensional TIs[55,56].

## Methods

**Circuit implementation and experimental procedure.** The implementation of the LC circuit, so as to map its impedance response to a target Hamiltonian, follows a design strategy similar to recent works, which targeted different topological models[20–23,26–28]. As explained in the main text, positive and negative hoppings in the tight-binding Hamiltonian are represented by capacitors and inductors respectively. Defining the complex conductance between sites $i$ and $j$ as $D_{ij} = i\alpha H_{ij}$, we take $\alpha = 2\pi f_0 C_0$ to map the positive nearest neighbour hopping $J = 1$ to capacitance $C_0 = 1$ nF, and the long-range hopping $J'$ to capacitance $C' = 2$ nF, at $f = f_0$. Next, setting $f_0 = 1/(2\pi\sqrt{L_0 C_0}) \approx 113$ kHz maps the negative nearest neighbour hopping to inductance $L_0 = 2$ mH, and the negative long-range hopping $J'' = -2$ to $L' = 1$ mH. Each site is connected to ground by additional components to satisfy Eq. (2); see Supplementary Note 2. The required capacitances are obtained by connecting 1 nH capacitors (Murata GCM155R71H102KA37D) in series or parallel, and the inductances are achieved by connecting 1 mH inductors (Taiyo Yuden LB2518T102K).

The circuit is divided into several printed circuit boards (PCBs), stacked on top of each other. Each PCB is divided into $6 \times 6 = 36$ nodes, corresponding to the dimensions of the 4D lattice in the $x$–$z$ plane (see Fig. 1c of the main text). Each $x$–$z$ lattice plane actually consists of several PCBs stacked with vertical electrical interconnects, in order to fit all the necessary circuit components.

We measure the impedance between any given node $r$ and the common ground by applying a 1 V sine wave of frequency $f_0$ on that node, and measuring the voltage $V_r$ and the current $I_r$. As stated in Eq. 4, the impedance between node $r$ and the ground is the $r$th diagonal term of the inverse of the circuit Laplacian $L$. Using Eq. (3), one obtains[20,27]

$$Z_r = \frac{i}{\alpha}\lim_{\epsilon \to 0}\sum_n \frac{|\psi_n(r)|^2}{E_n - E + i\epsilon},\qquad(6)$$

where $\psi_n(r)$ is the $n$-th energy eigenstate's amplitude on site $r$, and $E_n$ is the corresponding eigenenergy. Thus, if the impedance measurement is performed at $f = f_0$, then $\mathrm{Re}[Z_r] = (1/\pi\alpha)\sum_n\delta(E - E_n)|\psi_n(r)|^2$ is equivalent to the lattice's LDOS at energy $E$. With resistances present, the eigenenergies in Eq. (6) acquire an imaginary part, which has the effect of smoothing out the impedance curves (see Supplementary Note 3).

**Circuit simulations.** All circuit simulations are performed with ngspice, a free software circuit simulator. We assign to each 1 nF capacitor a 10 Ω resistance, consistent with the resistance in the manufacturer-supplied SPICE model at our operating frequency. For each 1 mH inductor, we assign a 24 Ω resistance consistent with the manufacturer-provided data sheet. Each resistance is applied in series with the corresponding capacitive or inductive element. Other sources of resistance, such as the PCB interconnects, are much harder to characterise and

were thus not accounted for in the circuit simulations. To model the disorder in the capacitors and inductors, we apply 10% uniformly-distributed disorder to each capacitance and inductance, consistent with the stated tolerances in their data sheets. The simulations are performed like the experiments: i.e. sine wave voltage source is applied to each node, and the steady-state voltage and current are used to determine the complex impedance.

## Data availability
The circuit measurement data that support the findings of this study are available in DR-NTU(data) with the identifier "https://doi.org/10.21979/N9/KXL3TD"[57].

## Code availability
Ngspice and Python code used for circuit simulation and generating all plots can be found in DR-NTU(data) with the identifier "https://doi.org/10.21979/N9/KXL3TD"[57].

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

## Acknowledgements

We are grateful to C.H. Lee and T. Ozawa for helpful discussions. This work was supported by the Singapore MOE Academic Research Fund Tier 3 Grant MOE2016-T3-1-006, Tier 1 Grants RG187/18 and RG174/16(S), and Tier 2 Grant MOE2018-T2-1-022(S). H.M.P. is supported by the Royal Society via grants UF160112, RGF/EA/180121 and RGF/R1/180071.

## Author contributions

Y.W. designed and fabricated the samples, and performed the experiments and simulations. B.Z. and Y.C. supervised the project. Y.W., H.M.P., B.Z. and Y.C. contributed to the theoretical analysis and the writing of the paper.

## Competing interest

The authors declare no competing interests.
