## [Peer Review File · Nature Communications]

Reviewers' comments:

Reviewer #1 (Remarks to the Author):

In this paper, the authors develop an electric circuit lattice with the goal of emulating a 4d topological insulator, class AI, and carry out the experimental characterization of 3d surface states. Numerical simulations and experimental results are consistent. However, there are several questions that need to be addressed before I may be able to conclude whether the paper may be feasible for publication in Nature Communications.

1) The authors claim that the circuit lattice represents a 4d topological insulator, however, the finite lattice they construct has only one single unit cell in two directions (in y and w). In such case, it is clear that this cannot map to the full spectrum of a 4d topological insulator, even though periodic conditions are applied in these directions. The reason is that the degrees of freedom are limited by the number of unit cells, thus the external degree of freedom in y and w direction is one in their system. As a consequence, the Hamiltonian of a 4d topological insulator in momentum representation cannot be mapped into this system, except at momentum $(0,0)$. In such circumstance, the topological surface states might be missed when they project Hamiltonian only at $(0,0)$ in momentum space, because it might happen that surface states can appear anywhere except at $(0,0)$. Luckily, this scenario doesn't happen because they indeed observe 3d surface states in their system. Of course, any finite system cannot disclose the true topology of an infinite TI, but I strongly suggest the authors enlarge their system along y and w directions and compare the new results with what they have now.

2). The authors should explicitly write down the TBM Hamiltonian that they map. It is also very informational to present the numerical band structures, including 3d surface dispersion, to prove the legality of their choice to target the Hamiltonian at $(0,0)$.

3). Can the authors explain the discrepancy between Fig. 2(a) and 2(b)? Why 3d surface states appear at larger parameter $|E|$ in 2(b) compared to TBM results in 2(a)? does the frequency-dependent Hamiltonian account for this difference? Or is it connected to the targeted Hamiltonian at $(0,0)$ raised up in question 1)?

4) I understand that the kernel Hamiltonian of the electric circuit maps to the desired TBM Hamiltonian only at resonant frequency f_0 , I worry whether such frequency dependence affects the topological invariant, or distorts the topological transition too much. The authors need to provide more investigation in this direction and comment on this issue.

5) The authors point out that the lattice consists of 4 sublattices, which indicates that the internal degree freedom of the unit cell is 4. However, they say later "We target a finite 4D lattice with three unit cells (6 sites) in the x and z directions, and one unit cell (2 sites) in y and w ." Can they explain why each unit cell has only 2 sites instead of 4 sites?

6) Details of the ground plane in the PCBs are missing, the authors should provide them either in the Methods section or in the caption of Fig. 1.

Reviewer #2 (Remarks to the Author):

The authors employ topoelectric circuits to realize a four-dimensional topological insulator. This subfield of synthetic topological matter, initiated by Refs 17,18 while crucially reinitialized by Ref. 20, has

recently witnessed an enormous interest from theory and experiment. As already explicitly anticipated by Ref. 20, the network character of topoelectric circuits readily allows the creation of any explicit connectivity, and as such in principle any dimensionality. The authors know the literature well, and have carefully referenced their work. Employing a four-dimensional network connectivity, the authors have realized a four-dimensional topoelectrical circuit, and measured its bulk and in particular its edge mode profile.

In terms of novelty, the paper, in my opinion, meets the criteria for Nature Communications. I recommend the manuscript for publication.

We are grateful to the reviewers for their well-informed and constructive feedback on the manuscript “Circuit Implementation of a Four-Dimensional Topological Insulator” (NCOMMS-20-04146-T). Our detailed responses to the reviewer comments are given below.

Response to Reviewer #1

1) The authors claim that the circuit lattice represents a 4d topological insulator, however, the finite lattice they construct has only one single unit cell in two directions (in y and w). In such case, it is clear that this cannot map to the full spectrum of a 4d topological insulator, even though periodic conditions are applied in these directions. The reason is that the degrees of freedom are limited by the number of unit cells, thus the external degree of freedom in y and w direction is one in their system. As a consequence, the Hamiltonian of a 4d topological insulator in momentum representation cannot be mapped into this system, except at momentum $(0,0)$. In such circumstance, the topological surface states might be missed when they project Hamiltonian only at $(0,0)$ in momentum space, because it might happen that surface states can appear anywhere except at $(0,0)$. Luckily, this scenario doesn't happen because they indeed observe 3d surface states in their system. Of course, any finite system cannot disclose the true topology of an infinite TI, but I strongly suggest the authors enlarge their system along y and w directions and compare the new results with what they have now.

We agree with reviewer's comment that applying periodic boundary conditions along y and w , with one lattice period, is equivalent to mapping to $k_y = k_w = 0$ in momentum space. The gap in the bulk spectrum of the tight-binding model closes at $k_y=k_w=0$ at the topological transition, as shown in subplot **a-c** below (see also arXiv:1806.05263), allowing the topological surface states to be efficiently sampled. In fact, we designed the experimental lattice with these facts in mind. In the revised manuscript, we have inserted a sentence to clarify this issue (see line 93), and we are grateful to the reviewer for raising this point. We have also plotted the surface state dispersion relations in the newly revised Supplementary Note 1 (reproduced in subplot **d,e** below).

As suggested by the referee, we have also double-checked the effects of finite system size along y and w by numerically calculating the states of finite lattices. The figure below shows the phase diagram

for lattices with different sizes (2 sites versus 6 sites) along the y and w directions. The results are qualitatively very similar, as expected. It is also worth highlighting that this would correspond experimentally to a huge increase in size, from 144 to 1296 lattice sites, with little benefit.

By contrast, as explained in the main text (especially Fig. 3 of the main text), if the lattice is enlarged along the open directions x and z , the spectrum becomes a progressively better match for the band diagram of the infinite lattice (including quantitatively matching the band edges). This is why we choose to put more unit cells along x and z rather than y and w in the experiment.

2). The authors should explicitly write down the TBM Hamiltonian that they map. It is also very informational to present the numerical band structures, including 3d surface dispersion, to prove the legality of their choice to target the Hamiltonian at (0,0).

We are grateful to the referee for this suggestion. In the new Supplementary Note 1, we have written down both the explicit real space tight-binding Hamiltonian and its k -space counterpart. We also give plots of the bulk dispersion relation, showing that the gap-closing occurs at $k_y = k_w = 0$, as well as the surface dispersion.

3). Can the authors explain the discrepancy between Fig. 2(a) and 2(b)? Why 3d surface states appear at larger parameter $|E|$ in 2(b) compared to TBM results in 2(a)? does the frequency-dependent Hamiltonian account for this difference? Or is it connected to the targeted Hamiltonian at (0,0) raised up in question 1)?

The 3d surface states appear at a larger parameter $|E|$ in Fig. 2(b) because of a finite size effect, specifically the finite size in the x and z directions where the lattice are truncated (Dirichlet boundary conditions) to 6 sites each. This is not due to the frequency dependence in the Hamiltonian, as Fig. 2(b) is calculated directly from the tight-binding model rather than the circuit equations. To verify our claim that this is a finite-size effect, Fig. 3 of the manuscript shows the spectrum for different lattices sizes in x and z . With progressively larger lattices, the spectrum becomes a better matches the infinite-lattice band diagram.

4) I understand that the kernel Hamiltonian of the electric circuit maps to the desired TBM Hamiltonian only at resonant frequency f_0 , I worry whether such frequency dependence affects the topological invariant, or distorts the topological transition too much. The authors need to provide more investigation in this direction and comment on this issue.

We thank the reviewer for raising this important issue. The key point is that the target tight-binding Hamiltonian has a topologically nontrivial bandgap at the reference energy E and frequency f_0 . If we tune away from this point, so long as (i) the gap remains open and (ii) the TI symmetry class remains unchanged, that gap remains topologically nontrivial and the topological invariant associated with that gap (in this case the second Chern number) remains the same. Point (i) is consistent with our experimental results, which show that the TI-like features (bulk gap and enhanced surface response) persist over a range of frequencies and m values around the reference point. Point (ii) holds because the effect of the frequency shift is to alter the negative hopping terms (which are realized by inductors), multiplying them by a factor f_0^2/f^2 ; the altered Hamiltonian continues to belong to Class AI.

We have revised the last paragraph of the “Experimental Results” section (page 6 onward) to explain the above points clearly: *“The frequency dependence of the circuit impedance is also consistent with the spectral features of a topological insulator at small values of m ... These experimental results are in good agreement with simulations (Fig. 4g–j).”*

Incidentally, the mapping of different frequencies to different Hamiltonians is not unique to the present circuit system. A similar feature was already present in the very first demonstration of a classical topological band insulator, namely the gyromagnetic photonic crystal described in PRL 100, 013905 (2008) and implemented in Nature 461, 772 (2009) by the MIT group. In that system, the gyromagnetic medium is dispersive (frequency dependent), and the implications for band topology are also accounted for in the manner described above.

5) The authors point out that the lattice consists of 4 sublattices, which indicates that the internal degree freedom of the unit cell is 4. However, they say later “We target a finite 4D lattice with three unit cells (6 sites) in the x and z directions, and one unit cell (2 sites) in y and w.” Can they explain why each unit cell has only 2 sites instead of 4 sites?

There are 4 sites per unit cell, but the number of sites per unit cell *in each direction* can be less than 4. To reduce the chances of confusion, we have changed the wording of that passage in the revised manuscript to “6 sites along the x and z directions, and 2 sites along y and w”.

6) Details of the ground plane in the PCBs are missing, the authors should provide them either in the Methods section or in the caption of Fig. 1.

Information about how the sites are connected to ground is given in Supplementary Note 3 of the revised paper (which used to be Supplementary Note 2 in the previous submission) – see the discussion below Eq. (S13). To emphasize this detail in the revised manuscript, in the Methods section we added the sentence: “Each site is connected to ground by additional components to satisfy Eq. (12); see

Supplementary Note 2". We have also updated Fig. 1 and its caption to mention the grounding components.

Response to Reviewer #2

The authors employ topolelectric circuits to realize a four-dimensional topological insulator. This subfield of synthetic topological matter, initiated by Refs 17,18 while crucially reinitialized by Ref. 20, has recently witnessed an enormous interest from theory and experiment. As already explicitly anticipated by Ref. 20, the network character of topolelectric circuits readily allows the creation of any explicit connectivity, and as such in principle any dimensionality. The authors know the literature well, and have carefully referenced their work. Employing a four-dimensional network connectivity, the authors have realized a four-dimensional topolelectrical circuit, and measured its bulk and in particular its edge mode profile.

In terms of novelty, the paper, in my opinion, meets the criteria for Nature Communications. I recommend the manuscript for publication.

We are grateful to the reviewer for his/her positive comments.

REVIEWERS' COMMENTS:

Reviewer #1 (Remarks to the Author):

The authors have addressed my previous concerns, and the paper can be published in Nature Communications.